# Songbirds can learn flexible contextual control over syllable sequencing

**Lena Veit**[†]*, **Lucas Y Tian**[‡], **Christian J Monroy Hernandez**, **Michael S Brainard**\*

Center for Integrative Neuroscience and Howard Hughes Medical Institute, University of California, San Francisco, San Francisco, United States

**Abstract** The flexible control of sequential behavior is a fundamental aspect of speech, enabling endless reordering of a limited set of learned vocal elements (syllables or words). Songbirds are phylogenetically distant from humans but share both the capacity for vocal learning and neural circuitry for vocal control that includes direct pallial-brainstem projections. Based on these similarities, we hypothesized that songbirds might likewise be able to learn flexible, moment-by-moment control over vocalizations. Here, we demonstrate that Bengalese finches (*Lonchura striata domestica*), which sing variable syllable sequences, can learn to rapidly modify the probability of specific sequences (e.g. 'ab-c' versus 'ab-d') in response to arbitrary visual cues. Moreover, once learned, this modulation of sequencing occurs immediately following changes in contextual cues and persists without external reinforcement. Our findings reveal a capacity in songbirds for learned contextual control over syllable sequencing that parallels human cognitive control over syllable sequencing in speech.

**\*For correspondence:**
lena.veit@uni-tuebingen.de (LV);
msb@phy.ucsf.edu (MSB)

**Present address:** [†]Institute for
Neurobiology, University of
Tübingen, Tübingen, Germany;
[‡]The Rockefeller University, New
York, United States

**Competing interests:** The
authors declare that no
competing interests exist.

**Reviewing editor:** Jesse H
Goldberg, Cornell University,
United States

## Introduction

A crucial aspect of the evolution of human speech is the development of flexible control over learned vocalizations (*Ackermann et al., 2014*; *Belyk and Brown, 2017*). Humans have unparalleled control over their vocal output, with a capacity to reorder a limited number of learned elements to produce an endless combination of vocal sequences that are appropriate for current contextual demands (*Hauser et al., 2002*). This cognitive control over vocal production is thought to rely on the direct innervation of brainstem and midbrain vocal networks by executive control structures in the frontal cortex, which have become more elaborate over the course of primate evolution (*Hage and Nieder, 2016*; *Simonyan and Horwitz, 2011*). However, because of the comparatively limited flexibility of vocal production in nonhuman primates (*Nieder and Mooney, 2020*), the evolutionary and neural circuit mechanisms that have enabled the development of this flexibility remain poorly understood.

Songbirds are phylogenetically distant from humans, but they have proven a powerful model for investigating neural mechanisms underlying learned vocal behavior. Song learning exhibits many parallels to human speech learning (*Doupe and Kuhl, 1999*); in particular, juveniles need to hear an adult tutor during a sensitive period, followed by a period of highly variable sensory-motor exploration and practice, during which auditory feedback is used to arrive at a precise imitation of the tutor song (*Brainard and Doupe, 2002*). This capacity for vocal learning is subserved by a well-understood network of telencephalic song control nuclei. Moreover, as in humans, this vocal control network includes strong projections directly from cortical (pallial) to brainstem vocal control centers (*Doupe and Kuhl, 1999*; *Simonyan and Horwitz, 2011*). These shared behavioral features and neural specializations raise the question of whether songbirds might also share the capacity to learn flexible control over syllable sequencing.

Contextual variation of song in natural settings, such as territorial counter-singing or female-directed courtship song, indicate that songbirds can rapidly alter aspects of their song, including

**eLife digest** Human speech and birdsong share numerous parallels. Both humans and birds learn their vocalizations during critical phases early in life, and both learn by imitating adults. Moreover, both humans and songbirds possess specific circuits in the brain that connect the forebrain to midbrain vocal centers.

Humans can flexibly control what they say and how by reordering a fixed set of syllables into endless combinations, an ability critical to human speech and language. Birdsongs also vary depending on their context, and melodies to seduce a mate will be different from aggressive songs to warn other males to stay away. However, so far it was unclear whether songbirds are also capable of modifying songs independent of social or other naturally relevant contexts.

To test whether birds can control their songs in a purposeful way, Veit et al. trained adult male Bengalese finches to change the sequence of their songs in response to random colored lights that had no natural meaning to the birds. A specific computer program was used to detect different variations on a theme that the bird naturally produced (for example, "ab-c" versus "ab-d"), and rewarded birds for singing one sequence when the light was yellow, and the other when it was green. Gradually, the finches learned to modify their songs and were able to switch between the appropriate sequences as soon as the light cues changed. This ability persisted for days, even without any further training.

This suggests that songbirds can learn to flexibly and purposefully modify the way in which they sequence the notes in their songs, in a manner that parallels how humans control syllable sequencing in speech. Moreover, birds can learn to do this 'on command' in response to an arbitrarily chosen signal, even if it is not something that would impact their song in nature.

Songbirds are an important model to study brain circuits involved in vocal learning. They are one of the few animals that, like humans, learn their vocalizations by imitating conspecifics. The finding that they can also flexibly control vocalizations may help shed light on the interactions between cognitive processing and sophisticated vocal learning abilities.

syllable sequencing and selection of song types (*Chen et al., 2016*; *Heinig et al., 2014*; *King and McGregor, 2016*; *Sakata et al., 2008*; *Searcy and Beecher, 2009*; *Trillo and Vehrencamp, 2005*). However, such modulation of song structure is often described as affectively controlled (*Berwick et al., 2011*; *Nieder and Mooney, 2020*). For example, the presence of potential mates or rivals elicits a global and unlearned modulation of song intensity (*James et al., 2018*) related to the singer's level of arousal or aggression (*Alcami et al., 2021*; *Heinig et al., 2014*; *Jaffe and Brainard, 2020*). Hence, while prior observations suggest that a variety of ethologically relevant factors can be integrated to influence song production in natural settings, it remains unclear whether song can be modified more flexibly by learned or cognitive factors.

Here, we tested whether Bengalese finches can learn to alter specifically targeted vocal sequences within their songs in response to arbitrarily chosen visual cues, independent of social or other natural contexts. Each Bengalese finch song repertoire includes ~5–12 acoustically distinct elements ('syllables') that are strung together into sequences in variable but non-random order. For a given bird, the relative probabilities of specific transitions between syllables normally remain constant over time (*Okanoya, 2004*; *Warren et al., 2012*), but previous work has shown that birds can gradually adjust the probabilities of alternative sequences in response to training that reinforces the production of some sequences over others. In this case, changes to syllable sequencing develop over a period of hours to days (*Warren et al., 2012*). In contrast, we investigate here whether birds can learn to change syllable sequencing on a moment-by-moment basis in response to arbitrary visual cues that signal which sequences are adaptive at any given time. Our findings reveal that songbirds can learn to immediately, flexibly, and adaptively adjust the sequencing of selected vocal elements in response to learned contextual cues.

## Results

### Bengalese finches can learn context-dependent syllable sequencing

For each bird in the study, we first identified variably produced syllable sequences that could be gradually modified using a previously described aversive reinforcement protocol ('single context training'; *Tumer and Brainard, 2007*; *Warren et al., 2012*). For example, a bird that normally transitioned from the fixed syllable sequence 'ab' to either 'c' or 'd' (*Figure 1A,B*, sequence probability of ~36% for 'ab-c' and ~64% for 'ab-d') was exposed to an aversive burst of white noise (WN) feedback immediately after the 'target sequence' 'ab-d' was sung. In response, the bird learned over a period of days to gradually decrease the relative probability of that sequence in favor of the alternative sequence 'ab-c' (*Figure 1C*). This change in sequence probabilities was adaptive in that it enabled the bird to escape from WN feedback. Likewise, when the sequence, 'ab-c' was targeted, the probability of 'ab-d' increased gradually over several days of training (*Figure 1D*). These examples are consistent with prior work that showed such sequence modifications develop over a period of several days, with the slow time course suggesting a gradual updating of synaptic connections within syllable control networks in response to performance-related feedback (*Warren et al., 2012*). In contrast, the ability to immediately and flexibly reorder vocal elements in speech must reflect mechanisms that enable contextual factors to exert moment-by-moment control over selection and sequencing of alternative vocal motor programs. Having identified sequences for each bird for which the probability of production could be gradually modified in this manner, we then tested whether birds could be trained to rapidly switch between those same sequences in a context-dependent manner.

To determine whether Bengalese finches can learn to flexibly select syllable sequences on a moment-by-moment basis, we paired WN targeting of specific sequences with distinct contextual cues. In this context-dependent training protocol, WN was targeted to defined sequences in the bird's song as before, but the specific target sequence varied across alternating blocks, signaled by different colored lights in the home cage (see Materials and methods). *Figure 1E* shows an example experiment, with 'ab-d' targeted in yellow light, and 'ab-c' in green light. At baseline, without WN, switches between yellow and green contexts (at random intervals of 0.5–1.5 hr) did not lead to significant changes in the relative proportion of the target sequences, indicating that there was no inherent influence of the light cues on sequence probabilities (*Figure 1F*, p(ab-d) in yellow vs. green context was 67 ± 1.6% vs. 64 ± 1.5%, p=0.17, rank-sum test, n = 53 context blocks from baseline period). Training was then initiated in which WN was alternately targeted to each sequence, over blocks that were signaled by light cues. After 2 weeks of such context-specific training, significant sequencing differences developed between light contexts that were appropriate to reduce aversive feedback in each context (*Figure 1G*, p(ab-d) in yellow vs. green context shifted to 36.5 ± 4.8% vs. 83.1 ± 3.5%, p<0.01, rank-sum test, n = 22 context blocks, block duration between 1 and 2.5 hr). Likewise, for all birds trained on this protocol (n = 8), context-dependent sequencing differences developed in the appropriate direction over a period of weeks (27 ± 6% difference in probabilities between contexts after a mean of 33 days training, versus 1% ± 2% average difference in probabilities at baseline; p<0.01, n = 8, signed rank test, *Figure 1H*). Thus, Bengalese finches are able to learn context-specific modifications to syllable sequencing.

### Syllable sequencing shifts immediately following switches in context

Contextual differences between different blocks could arise through an immediate shift in sequence probabilities upon entry into a new context and/or by rapid learning within each block. We examined whether trained birds exhibited any immediate shifts in their syllable sequencing when entering a new light context by computing the average probability of target sequences across songs aligned with the switch between contexts (*Figure 2A,B*, example experiment). This 'switch-triggered average' revealed that across all birds, switches to the yellow context were accompanied by an immediate decrease in the probability of the yellow target sequence, whereas switches out of the yellow context (and into the green context) led to an immediate increase in the yellow target sequence (*Figure 2C,D*, p<0.05, signed rank test comparing first and last song, n = 8). To quantify the size of these immediate shifts, we calculated the difference in sequence probability from the last five songs in the previous context to the first five songs in the current context; this difference averaged

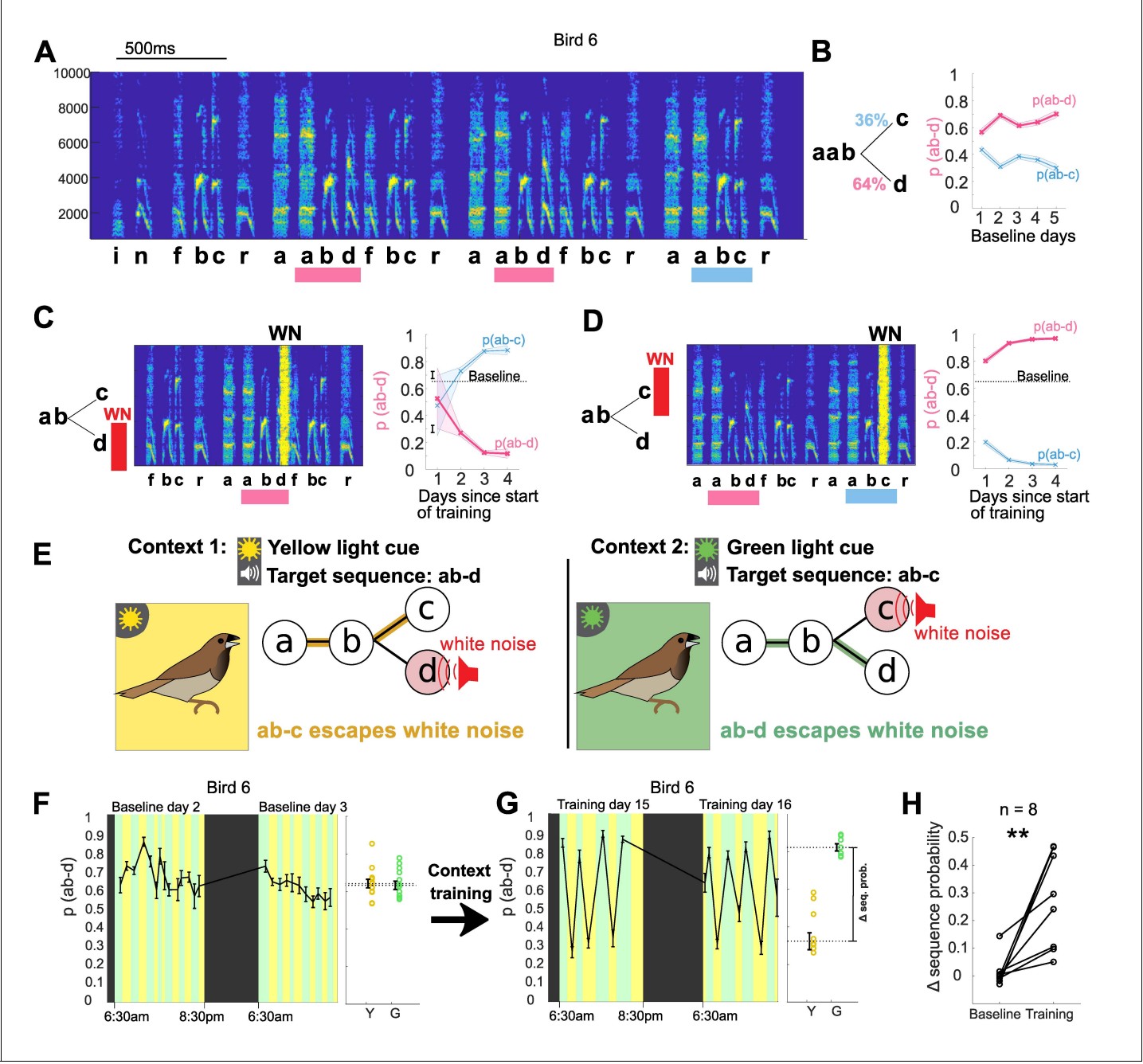

**Figure 1.** Bengalese finches can learn context-dependent sequencing. (**A**) Example spectrogram highlighting points in song with variable sequencing. Syllables are labeled based on their spectral structure, target sequences for the different experiments (ab-c and ab-d) are marked with colored bars. Y-axis shows frequency in Hz. (**B**) Transition diagram with probabilities for sequences ab-c and ab-d. The sequence probability of ab-d (and complementary probability ab-c) stayed relatively constant over five days. Shaded area shows 95% confidence interval for sequence probability. Source data in *Figure 1—source data 3*. (**C**) Aversive reinforcement training. Schematic showing aversive WN after target sequence ab-d; spectrogram shows WN stimulus, covering part of syllable d. WN targeted to sequence ab-d led to a gradual decrease in the probability of that sequence over several days, and a complementary increase in the probability of ab-c. (**D**) WN targeted to ab-c led to a gradual increase in the sequence probability of ab-d. Source data in *Figure 1—source data 2*. (**E**) Schematic of the contextual learning protocol, with target for WN signaled by colored lights. (**F**) Left: Two example days of baseline without WN but with alternating blocks of green and yellow context. Colors indicate light context (black indicates periods of lights off during the night), error bars indicate SEM across song bouts in each block. Right: Average sequence probability in yellow and green blocks during baseline. Open circles show individual blocks, error bars show SEM across blocks. (**G**) Left: Two example days after training (WN on). Right: Average sequence probability in yellow and green blocks after training. (**H**) Contextual difference in sequence probability for eight trained birds before and after training (\*\*p<0.01 signed rank test). Source data in *Figure 1—source data 1*.

*Figure 1 continued*

The online version of this article includes the following source data for figure 1:

**Source data 1.** Switch magnitude during baseline and after training for all birds, to generate *Figure 1H*, and plots like *Figure 1F,G* for all birds.
**Source data 2.** Sequence data for the example bird during single-context training, to generate *Figure 1C,D*.
**Source data 3.** Sequence data for the example bird during baseline, to generate *Figure 1B*.

---

0.24 ± 0.06 for switches to green light and −0.22 ± 0.06 for switches to yellow light (*Figure 2E,F*). These results indicate that birds could learn to immediately recall an acquired memory of context-appropriate sequencing upon entry into each context, even before having the chance to learn from reinforcing feedback within that context.

We next asked whether training additionally led to an increased rate of learning within each context, which also might contribute to increased contextual differences over time. Indeed, such faster re-learning for consecutive encounters of the same training context, or 'savings', is sometimes observed in contextual motor adaptation experiments (*Lee and Schweighofer, 2009*). To compare the magnitude of the immediate shift and the magnitude of within-block learning over the course of training, we plotted the switch-aligned sequence probabilities at different points in the training process. *Figure 2G* shows for the example bird that the magnitude of the shift (computed between the first and last five songs across context switches) gradually increased over 11 days of training. *Figure 2H* shows the switch-aligned sequence probability trajectories (as in *Figure 2A,B*) for this bird early in training (red) and late in training (blue), binned into groups of seven context switches. Qualitatively, there was both an abrupt change in sequence probability at the onset of each block (immediate shift at time point 0) and a gradual adjustment of sequence probability within each block (within-block learning over the first 80 songs following light switch). Over the course of training, the immediate shift at the onset of each block got larger, while the gradual change within blocks stayed approximately the same (learning trajectories remained parallel over training, *Figure 2H*). Linear fits to the sequence probabilities for each learning trajectory (i.e. the right side of *Figure 2H*) reveal that, indeed, the change in sequence probability at the onset of blocks (i.e. intercepts) increased over the training process (*Figure 2K*), while the rate of change within blocks (i.e. slopes) stayed constant (*Figure 2I*). To quantify this across birds, we measured the change over the course of learning in both the magnitude of immediate shifts (estimated as the intercepts from linear fits) and the rate of within-block learning (estimated as the slopes from linear fits). As for the example bird, we found that the rate of learning within each block stayed constant over time for all five birds (*Figure 2L*). In contrast, the magnitude of immediate shifts increased over time for all birds (*Figure 2L*). These analyses indicate that adjustments to sequence probability reflect two dissociable processes, an immediate cue-dependent shift in sequence probability at the beginning of blocks, that increases with contextual training, and a gradual adaptation of sequence probability within blocks, that does not increase with contextual training.

## Visual cues in the absence of reinforcement are sufficient to evoke sequencing changes

The ability of Bengalese finches to implement an immediate shift in sequencing on the first rendition in a block – and thus before they have a chance to learn from reinforcing feedback – argues that they can maintain context-specific motor memories and use contextual visual cues to anticipate correct sequencing in each context. To explicitly test whether birds can flexibly switch between sequencing appropriate for distinct contexts using only visual cues, we included short probe blocks which presented the same light cues without WN stimulation. Probe blocks were interspersed in the sequence of training blocks so that each switch between types of blocks was possible and, on average, every third switch was into a probe block (see Materials and methods). Light switches into probe blocks were associated with similar magnitude shifts in sequence probability as switches into WN blocks of the corresponding color (−0.22 ± 0.06 to both yellow WN and yellow probe blocks from green WN blocks, p=0.94, signed rank test; 0.24 ± 0.06 to green WN and 0.23 ± 0.07 to green probe blocks from yellow WN blocks, p=0.64, signed rank test). As the most direct test of whether light cues alone evoke adaptive sequencing changes, we compared songs immediately before and

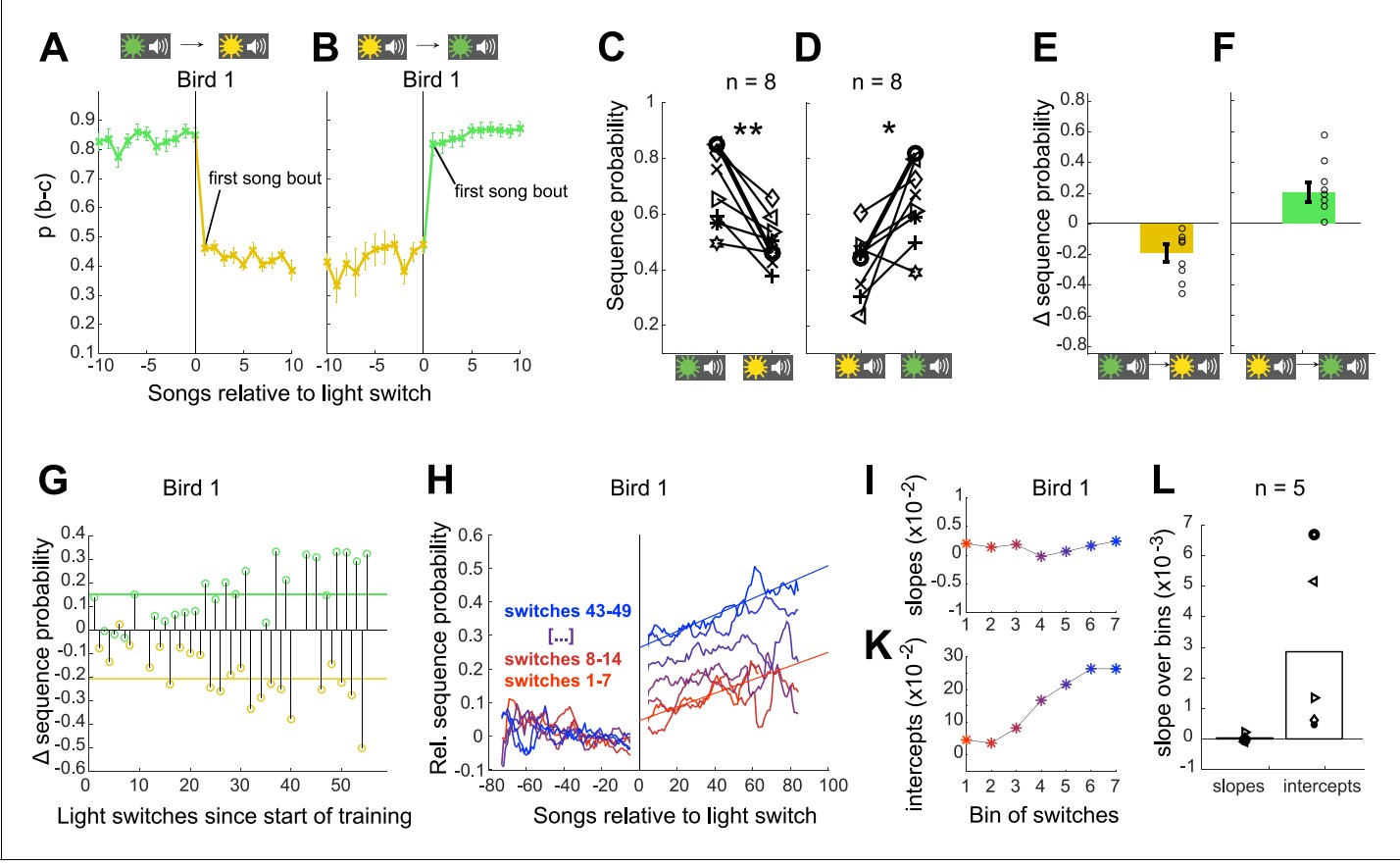

**Figure 2.** Sequence probabilities shift immediately following a switch in context. (A, B) Average sequence probability per song for example Bird 1 aligned to switches from green to yellow context (A) and from yellow to green context (B). Error bars indicate SEM across song bouts (n = 35 switches (A), n = 33 switches (B)). (C) Changes in sequence probability from the last song in green context to the first song in yellow context for all eight birds. Example bird in (A, B) highlighted in bold. **p<0.01 signed rank test. (D) Changes in sequence probability from the last song in yellow context to the first song in green context. *p<0.05 signed rank test. (E) Shift magnitudes for all birds, defined as the changes in sequence probability from the last five songs in the green context to the first five songs in the yellow context. Open circles show individual birds, error bars indicate SEM across birds. (F) Same as (E) for switches from yellow to green. Source data in *Figure 2—source data 1*. (G) Shift magnitudes over training time for the example bird (11 days and 49 context switches; seven of the original 56 context switches are excluded from calculations of shift magnitudes because at least one of the involved blocks contained only one or two song bouts.). (H) Trajectory of switch-aligned sequence probabilities for the example bird early in training (red) and late in training (blue). Probabilities are normalized by the sequence probability in preceding block, and plotted so that the adaptive direction is positive for both switch directions (i.e. inverting the probabilities for switches to yellow.) (I) Slopes of fits to the sequence probability trajectories over song bouts within block. Units in change of relative sequence probability per song bout. (K) Intercepts of fits to sequence probability trajectories over song bouts within block. Units in relative sequence probability. (L) Changes in slopes and changes in intercepts for five birds over the training process, determined as the slopes of linear fits to curves as in (I and K) for each bird. Source data in *Figure 2—source data 2*.

The online version of this article includes the following source data for figure 2:

**Source data 1.** Switch magnitude between all contexts after training, to generate *Figures 2C–F* and *3E–H*.
**Source data 2.** Summary of training data, to generate *Figure 2L*.

after switches between probe blocks without intervening WN training blocks (probe-probe switches). *Figure 3A,B* shows song bouts for one example bird (Bird 2) which were sung consecutively across a switch from yellow probe to green probe blocks. In the first song following the probe-probe switch, the yellow target sequence ('f-ab') was more prevalent, and the green target sequence ('n-ab') was less prevalent, and such an immediate effect was also apparent in the average sequence probabilities for this bird aligned to probe–probe switches (*Figure 3C,D*). Similar immediate and appropriately directed shifts in sequencing at switches between probe blocks were observed for all eight birds (*Figure 3E,F*, p<0.05 signed rank test, n = 8), with average shifts in sequence probabilities of −0.21 ± 0.09 and 0.17 ± 0.08 (*Figure 3G,H*). The presence of such changes in the first songs sung

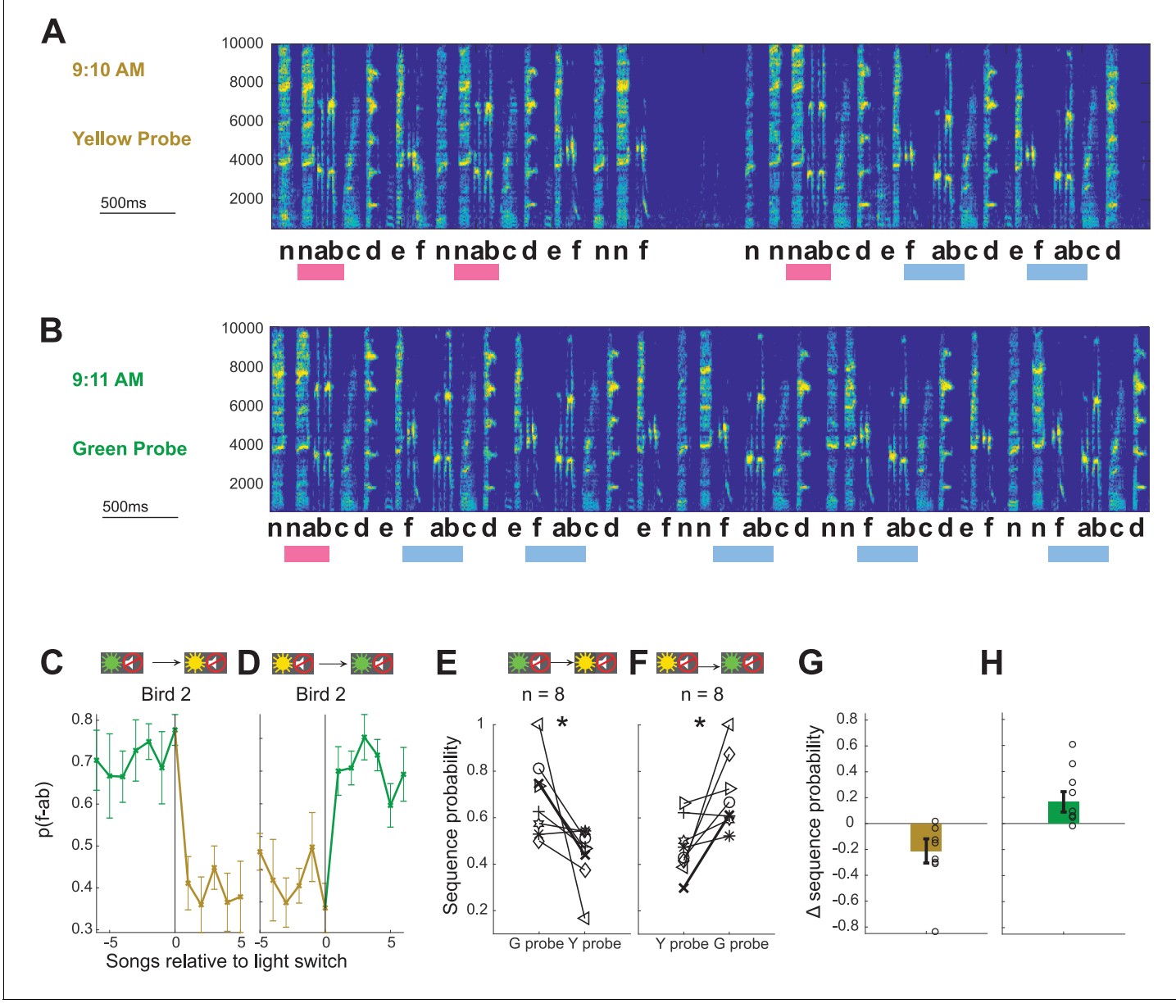

**Figure 3.** Contextual cues alone are sufficient to enable immediate shifts in syllable sequencing. (A,B) Examples of songs sung by Bird 2 immediately before (A) and after (B) a switch from a yellow probe block to a green probe block (full song bouts in *Figure 3—figure supplement 1*). Scale for x-axis is 500 ms; y-axis shows frequency in Hz. (C, D) Average sequence probability per song for Bird 2 aligned to switches from green probe to yellow probe blocks (C) and from yellow probe to green probe blocks (D). Error bars indicate SEM across song bouts (n = 14 switches (C), 11 switches (D)). (E, F) Average sequence probabilities for all eight birds at the switch from the last song in green probe context and the first song in yellow probe context, and vice versa. Example Bird 2 is shown in bold. *p<0.05 signed rank test. (G, H) Shift magnitudes for probe–probe switches for all birds. Open circles show individual birds; error bars indicate SEM across birds. Source data in *Figure 2—source data 1*.

The online version of this article includes the following figure supplement(s) for figure 3:

**Figure supplement 1.** Example song bouts surrounding a probe–probe context switch.

after probe–probe switches indicates that visual cues alone are sufficient to cause anticipatory, learned shifts between syllable sequences.

## Contextual changes are specific to target sequences

A decrease in the probability of a target sequence in response to contextual cues must reflect changes in the probabilities of transitions leading up to the target sequence. However, such changes could be restricted to the transitions that immediately precede the target sequence, or alternatively could affect other transitions throughout the song. For example, for the experiment illustrated in *Figure 1*, the prevalence of the target sequence 'ab-d' was appropriately decreased in the yellow context, in which it was targeted. The complete transition diagram and corresponding transition matrix for this bird (*Figure 4A,B*) reveal that there were four distinct branch points at which syllables were variably sequenced (after 'cr', 'wr', 'i', and 'aab'). Therefore, the decrease in the target sequence 'ab-d' could have resulted exclusively from an increase in the probability of the alternative transition 'ab-c' at the branch point following 'aab'. However, a reduction in the prevalence of the target sequence could also have been achieved by changes in the probability of transitions earlier in song such that the sequence 'aab' was sung less frequently. To investigate the extent to which contextual changes in probability were specific to transitions immediately preceding target sequences, we calculated the difference between transition matrices in the yellow and green probe contexts (*Figure 4C*). This difference matrix indicates that changes to transition probabilities were highly specific to the branch point immediately preceding the target sequences (specificity was defined as the proportion of total changes which could be attributed to the branch points immediately preceding target sequences; specificity for branch point 'aab' was 83.2%). Such specificity to branch points that immediately precede target sequences was typical across experiments, including cases in which different branch points preceded each target sequence (*Figure 4D–F*, specificity 96.9%). Across all eight experiments, the median specificity of changes to the most proximal branch points was 84.95%, and only one bird, which was also the worst learner in the contextual training paradigm, had a specificity of less than 50% (*Figure 4G*). Hence, contextual changes were specific to target sequences and did not reflect the kind of global sequencing changes that characterize innate social modulation of song structure (*Sakata et al., 2008*; *Sossinka and Böhner, 1980*).

## Distinct sequence probabilities are specifically associated with different visual cues

Our experiments establish that birds can shift between two distinct sequencing states in response to contextual cues. In order to test whether birds were capable of learning to shift to these two states from a third neutral context, we trained a subset of three birds with three different color-cued contexts. For these birds, after completion of training with WN targeted to distinct sequences in yellow and green contexts (as described above), we introduced interleaved blocks cued by white light in which there was no reinforcement. After this additional training, switches from the unreinforced context elicited changes in opposite directions for the green and yellow contexts (example bird *Figure 5A*). All birds (n = 3) showed adaptive sequencing changes for the first song bout in probe blocks (*Figure 5B,C*) as well as immediate shifts in the adaptive directions for all color contexts (*Figure 5D*, 0.11 ± 0.04 and 0.19 ± 0.05 for switches to green WN and green probe blocks, respectively; −0.15 ± 0.06 and −0.09 ± 0.02 for switches to yellow WN and yellow probe blocks, respectively). While additional data would be required to establish the number of distinct associations between contexts and sequencing states that can be learned, these findings suggest that birds can maintain at least two distinct sequencing states separate from a 'neutral' state and use specific associations between cue colors and sequencing states to rapidly shift sequencing in distinct directions for each context.

## Discussion

Speech, thought, and many other behaviors are composed of ordered sequences of simpler elements. The flexible control of sequencing is thus a fundamental aspect of cognition and motor function (*Aldridge and Berridge, 2002*; *Jin and Costa, 2015*; *Tanji, 2001*). While the flexibility of human speech is unrivaled, our contextual training paradigm revealed a simpler, parallel capacity in birds to produce distinct vocal sequences in response to arbitrary contextual cues. The colors of the cues had no prior relevance to the birds, so that their meaning had to be learned as a new association between cues and the specific vocal sequences that were contextually appropriate (i.e. that

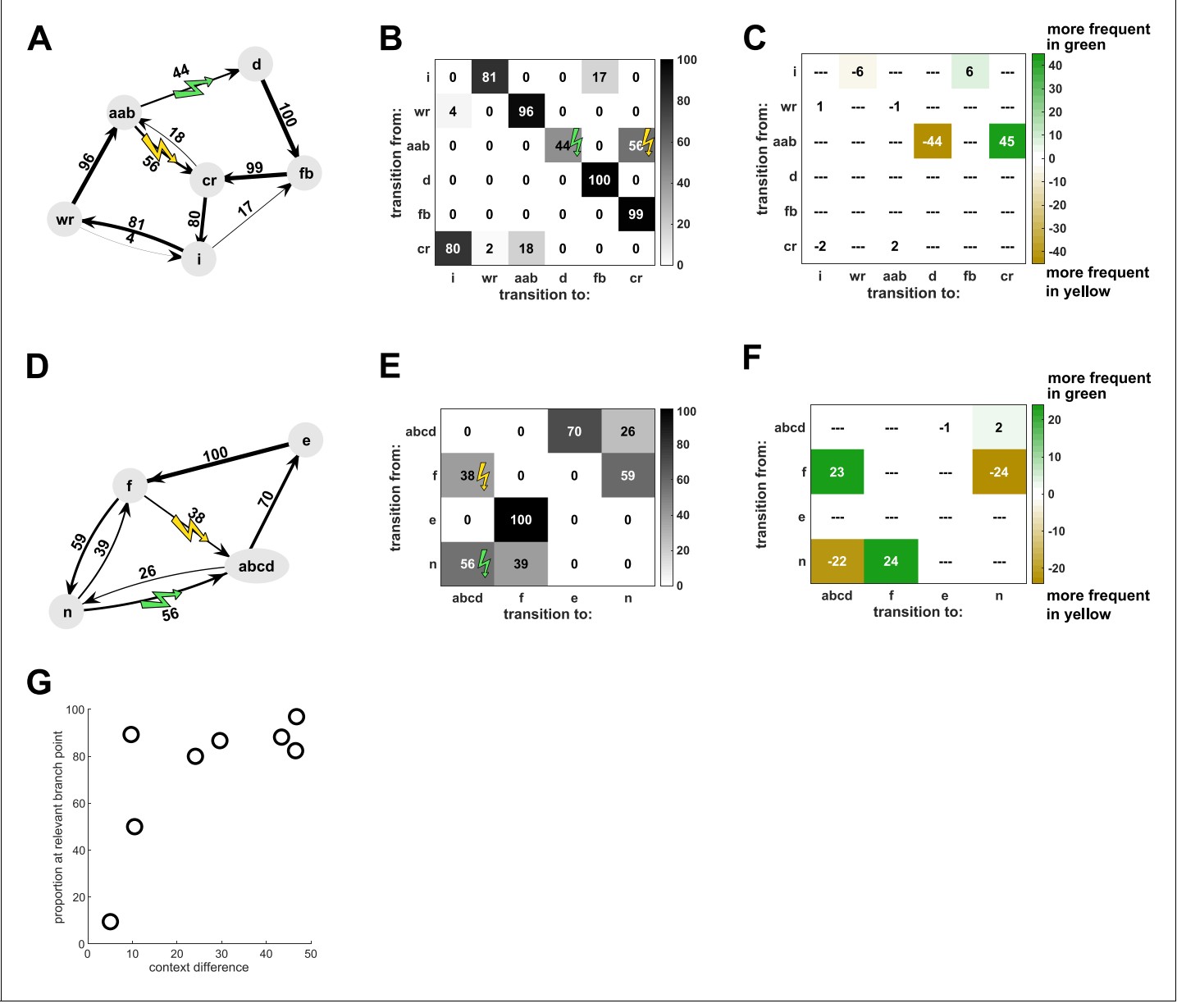

**Figure 4.** Contextual changes are local to the target sequences. (**A**) Transition diagram for the song of Bird 6 (spectrogram in *Figure 1*) in yellow probe context. Sequences of syllables with fixed transition patterns (e.g. 'aab') as well as repeat phrases and introductory notes have been summarized as single states to simplify the diagram. (**B**) Transition matrix for the same bird, showing same data as in (**A**). (**C**) Differences between the two contexts are illustrated by subtracting the transition matrix in the yellow context from the one in the green context, so that sequence transitions which are more frequent in green context are positive (colored green) and sequence transitions which are more frequent in yellow are negative (colored yellow). For this bird, the majority of contextual differences occurred at the branch point ('aab') which most closely preceded the target sequences ('ab-c' and 'ab-d'), while very little contextual difference occurred at the other three branch points ('i', 'wr', 'cr'). (**D–F**) Same for Bird 2 for which two different branch points ('f' and 'n') preceded the target sequences ('f-abcd' and 'n-abcd') (spectrogram in *Figure 3*). (**G**) Proportion of changes at the branch point(s) most closely preceding the target sequences, relative to the total magnitude of context differences for each bird (see Materials and methods). Most birds exhibited high specificity of contextual changes to the relevant branch points. Source data in *Figure 4—source data 1*.

The online version of this article includes the following source data and figure supplement(s) for figure 4:

**Source data 1.** Overview of different experimental parameters and song features for each bird, to generate (*Figure 4G*, *Figure 4—figure supplement 1*).
**Figure supplement 1.** Possible explanations for differences in contextual learning.

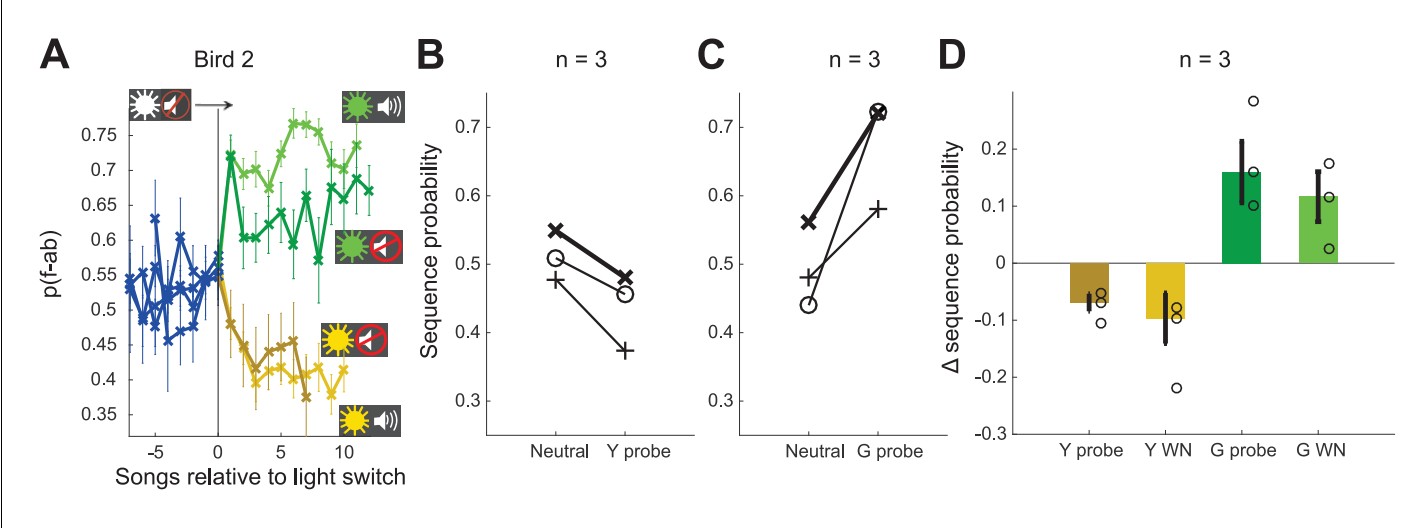

**Figure 5.** Contextual cues allow shifts in both directions. (A) Sequence probability for Bird 2 at the switch from neutral context to yellow and green WN contexts, as well as yellow and green probe contexts (no WN). Error bars indicate SEM across song bouts (n = 68 switches [green WN], 78 switches [yellow WN], 27 switches [green probe], 24 switches [yellow probe]). (B, C) Sequence probabilities for three birds for the last song in neutral context and the first song in the following probe context. Example bird in (A) highlighted in bold. (D) Shift magnitude for three birds at the switch from neutral context to all other contexts. Open circles show individual birds; error bars indicate SEM across birds. Source data in *Figure 5—source data 1*. The online version of this article includes the following source data for figure 5:

**Source data 1.** Switch magnitude during third context experiment, to generate *Figure 5B–D*.

escaped WN, given the current cues). Learned modulation of sequencing was immediately expressed in response to changes in cues, persisted following termination of training, and was largely restricted to the targeted sequences, without gross modifications of global song structure. Hence, for song, like speech, the ordering of vocal elements can be rapidly and specifically reconfigured to achieve learned, contextually appropriate goals. This shared capacity for moment-by-moment control of vocal sequencing in humans and songbirds suggests that the avian song system could be an excellent model for investigating how neural circuits enable flexible and adaptive reconfiguration of motor output in response to different cognitive demands.

## Flexible control of vocalizations

Our demonstration of contextual control over the ordering of vocal elements in the songbird builds on previous work showing that a variety of animals can learn to emit or withhold innate vocalizations in response to environmental or experimentally imposed cues. For example, nonhuman primates and other animals can produce alarm calls that are innate in their acoustic structure, but that are deployed in a contextually appropriate fashion (*Nieder and Mooney, 2020*; *Suzuki and Zuberbühler, 2019*; *Wheeler and Fischer, 2012*). Similarly, animals, including birds, can be trained to control their vocalizations in an experimental setting, by reinforcing the production of innate vocalizations in response to arbitrary cues to obtain food or water rewards (*Brecht et al., 2019*; *Hage and Nieder, 2013*; *Nieder and Mooney, 2020*; *Reichmuth and Casey, 2014*). In relation to these prior findings, our results demonstrate a capacity to flexibly reorganize the sequencing of learned vocal elements, rather than select from a fixed set of innate vocalizations, in response to arbitrary cues. This ability to contextually control the ordering, or syntax, of specifically targeted syllable transitions within the overall structure of learned song parallels the human capacity to differentially sequence a fixed set of syllables in speech.

The ability to alter syllable sequencing in a flexible fashion also contrasts with prior studies that have demonstrated modulation of vocalizations in more naturalistic settings. For example, songs produced in the context of courtship and territorial or aggressive encounters ('directed song') differ in acoustic structure from songs produced in isolation ('undirected song') (*Sakata et al., 2008*; *Searcy and Beecher, 2009*). This modulation of song structure by social context is characterized by global changes to the intensity of song production, with directed songs exhibiting faster tempo, and

greater stereotypy of both syllable structure and syllable sequencing, than undirected songs (*Sakata et al., 2008*; *Searcy and Beecher, 2009*; *Sossinka and Böhner, 1980*). This and other ethologically relevant modulation of song intensity may serve to communicate the singer's affective state, such as level of arousal or aggression (*Alcami et al., 2021*; *Hedley et al., 2017*; *Heinig et al., 2014*), and may largely reflect innate mechanisms (*James et al., 2018*; *Kojima and Doupe, 2011*) mediated by hypothalamic and neuromodulatory inputs to premotor regions (*Berwick et al., 2011*; *Gadagkar et al., 2019*; *James et al., 2018*; *Nieder and Mooney, 2020*). In contrast, here we show that birds can learn to locally modulate specific features of their songs (i.e. individually targeted syllable transitions) in response to arbitrarily assigned contextual cues that have no prior ethological relevance.

## Evolution of control over vocal sequencing

The capacity for moment-by-moment adjustment of vocalizations in response to arbitrary learned cues may depend on similar capacities that evolved to enable appropriate modulation of vocalizations in ethologically relevant natural contexts. For example, some species of songbirds preferentially sing different song types depending on factors such as time of day, location of the singer, or the presence of an audience (*Alcami et al., 2021*; *Hedley et al., 2017*; *King and McGregor, 2016*; *Searcy and Beecher, 2009*; *Trillo and Vehrencamp, 2005*). Even birds with only a single song type, such as Bengalese finches, vary parameters of their song depending on social context, including the specific identity of the listener (*Chen et al., 2016*; *Heinig et al., 2014*; *Sakata et al., 2008*). The ability to contextually control vocalizations is also relevant for the customization of vocal signatures for purposes of individual and group recognition (*Vignal et al., 2004*) and to avoid overlap and enhance communication during vocal turn-taking and in response to environmental noises (*Benichov and Vallentin, 2020*; *Brumm and Zollinger, 2013*). Such capacities for vocal control likely reflect evolutionary advantages of incorporating sensory and contextual information about conspecifics and the environment in generating increasingly sophisticated vocal signaling. Our results indicate a latent capacity to integrate arbitrary sensory signals into the adaptive deployment of vocalizations in songbirds and suggest that some of the contextual control observed in natural settings may likewise rely on learned associations and other cognitive factors. Perhaps evolutionary pressures to develop nuanced social communication led to the elaboration of cortical (pallial) control over brainstem vocal circuitry (*Hage and Nieder, 2016*), and thereby established a conduit that facilitated the integration of progressively more abstract cues and internal states in that control.

## Neural implementation of context-dependent vocal motor sequencing

The ability of birds to switch between distinct motor programs using visual cues is reminiscent of contextual speech and motor control studies in humans. For example, human subjects in both laboratory studies and natural settings can learn multiple 'states' of vocal motor adaptation and rapidly switch between them using contextual information (*Houde and Jordan, 2002*; *Keough and Jones, 2011*; *Rochet-Capellan and Ostry, 2011*). Similarly, subjects can learn two separate states of motor adaptation for other motor skills, such as reaching, and switch between them using cues or other cognitive strategies (*Cunningham and Welch, 1994*). Models of such context-dependent motor adaptation frequently assume at least two parallel processes (*Abrahamse et al., 2013*; *Ashe et al., 2006*; *Green and Abutalebi, 2013*; *Hikosaka et al., 1999*; *Lee and Schweighofer, 2009*; *McDougle et al., 2016*; *Rochet-Capellan and Ostry, 2011*; *Wolpert et al., 2011*), one that is more flexible, and sensitive to contextual information (*McDougle et al., 2016*), and a second that cannot readily be associated with contextual cues and is only gradually updated during motor adaptation (*Howard et al., 2013*). Specifically, in support of such a two-process model, *Imamizu and Kawato, 2009* and *Imamizu et al., 2007* found that contextual information can drive rapid shifts in adaptation at the beginning of new blocks, without affecting the rate of adaptation within blocks. The similar separation in our study between rapid context-dependent shifts in sequence probability at the onset of blocks, and gradual adaptation within blocks that does not improve with training (*Figure 2G–L*), suggests that such contextual sequence learning in the Bengalese finch may also be enabled by two distinct processes.

Humans studies of two-process models suggest that slow adaptation occurs primarily within primary motor structures, while fast context-dependent state switches, including for cued switching

between languages in bilinguals, engage more frontal areas involved in executive control (*Bialystok, 2017*; *Blanco-Elorrieta and Pylkkänen, 2016*; *De Baene et al., 2015*; *Imamizu and Kawato, 2009*). In songbirds, the gradual adaptation of sequence probabilities within blocks might likewise be controlled by motor and premotor song control structures, while visual contextual cues could be processed in avian structures analogous to mammalian prefrontal cortex, outside the song system. For example, the association area nidopallium caudolaterale (*Güntürkün, 2005*), is activated by arbitrary visual cues that encode learned rules (*Veit and Nieder, 2013*; *Veit et al., 2015*), and this or other avian association areas (*Jarvis et al., 2013*) may serve as an intermediate representation of the arbitrary contextual cues that can drive rapid learned shifts in syllable sequencing.

At the level of song motor control, our results indicate a greater capacity for rapid and flexible adjustment of syllable transition probabilities than previously appreciated. Current models of song production include networks of neurons in the vocal premotor nucleus HVC responsible for the temporal control of individual syllables, which are linked together by activity in a recurrent loop through brainstem vocal centers (*Andalman et al., 2011*; *Ashmore et al., 2005*; *Cohen et al., 2020*; *Hamaguchi et al., 2016*). At branch points in songs with variable syllable sequencing, one influential model posits that which syllable follows a branch point is determined by stochastic processes that depend on the strength of the connections between alternative syllable production networks, and thus dynamics local to HVC (*Jin, 2009*; *Jin and Kozhevnikov, 2011*; *Troyer et al., 2017*; *Zhang et al., 2017*). Such models could account for a gradual adjustment of sequence probabilities over a period of hours or days (*Lipkind et al., 2013*; *Warren et al., 2012*) through plasticity of motor control parameters, such as the strength of synaptic connections within HVC. However, our results demonstrate that there is not a single set of relatively fixed transition probabilities that undergo gradual adjustments, as could be captured in synaptic connectivity of branched syllable control networks. Rather, the song system has the capacity to maintain distinct representations of transition probabilities and can immediately switch between those in response to visual cues. HVC receives a variety of inputs that potentially could convey such visual or cognitive influences on sequencing (*Bischof and Engelage, 1985*; *Cynx, 1990*; *Seki et al., 2008*; *Ullrich et al., 2016*; *Wild, 1994*), and one of these inputs, Nif, has previously been shown to be relevant for sequencing (*Hosino and Okanoya, 2000*; *Vyssotski et al., 2016*). It therefore is likely that the control of syllable sequence in Bengalese finches involves a mix of processes local to nuclei of the song motor pathway (*Basista et al., 2014*; *Zhang et al., 2017*) as well as inputs that convey a variety of sensory feedback and contextual information. The well-understood circuitry of the avian song system makes this an attractive model to investigate how such top-down pathways orchestrate the kind of contextual control of vocalizations demonstrated in this study, and more broadly to uncover how differing cognitive demands can flexibly and adaptively reconfigure motor output.

## Materials and methods

### Subjects and sound recordings

The experiments were carried out on eight adult male Bengalese finches (*Lonchura striata*) obtained from the lab's breeding colony (age range 128–320 days post-hatch, median 178 days, at start of experiment). Birds were placed in individual sound-attenuating boxes with continuous monitoring and auditory recording of song. Song was recorded using an omnidirectional microphone above the cage. We used custom software for the online recognition of target syllables and real-time delivery of short 40 ms bursts of WN depending on the syllable sequence (*Tumer and Brainard, 2007*; *Warren et al., 2012*). This LabView program, EvTAF, is included as an executable file with this submission, and further support is available from the corresponding authors upon request. All procedures were performed in accordance with animal care protocols approved by the University of California, San Francisco Institutional Animal Care and Use Committee (IACUC).

### Training procedure and blocks

Bengalese finch song consists of a discrete number of vocal elements, called syllables, that are separated by periods of silence. At the start of each experiment, a template was generated to recognize a specific sequence of syllables (the target sequence) for each bird based on their unique spectral structure. In the context-dependent auditory feedback protocol, the target sequence that received

aversive WN feedback switched between blocks of different light contexts. Colored LEDs (superbrightleds.com, St. Louis, MO; green 520 nm, amber 600 nm) produced two visually distinct environments (green and yellow) to serve as contextual cues to indicate which sequences would elicit WN and which would 'escape' (i.e. not trigger WN). We wanted to test whether the birds would be able to associate song changes with any arbitrary visual stimulus; therefore, there was no reason to choose these specific colors, and the birds' color perception in this range should not matter, as long as they were able to discriminate the colors. The entire day was used for data acquisition by alternating the two possible light contexts. We determined sensitivity and specificity of the template to the target sequence on a randomly selected set of 20 song bouts on which labels and delivery of WN was hand-checked. Template sensitivity was defined as follows: sensitivity = (number of correct hits)/ (total number of target sequences). The average template sensitivity across experiments was 91.3% (range 75.2–100%). Template specificity was defined as: specificity = (number of correct escapes)/ (number of correct escapes plus number of false alarms), where correct escapes were defined as the number of target sequences of the currently inactive context that were not hit by WN, and false alarms were defined as any WN that was delivered either on the target sequence of the currently inactive context, or anywhere else in song. The average template specificity was 96.7% (range 90.6–100%).

At the start of each experiment, before WN training, songs were recorded during a baseline period in which cage illumination was switched between colors at random intervals. Songs from this baseline period were separately analyzed for each light color to confirm that there was no systematic, unlearned effect of light cues on sequencing before training. During initial training, cage illumination was alternatingly switched between colors at random intervals. Intervals were drawn from uniform distributions which differed between birds (60–150 min [four birds], 10–30 min [two birds], 60–240 min [one bird], 30–150 min [one bird]). Different training schedules were assigned to birds arbitrarily and were not related to a bird's performance. After an extended period of training (average 33 days, range 12–79 days), probe blocks without WN were included, to test whether sequencing changes could be elicited by visual cues alone. During this period, probe blocks were interspersed with WN training blocks. Probe blocks made up approximately one third of total blocks (10 of 34 blocks in the sequence) and 7–35% of total time, depending on the bird. The duration of probe blocks was typically shorter or equal to the duration of WN blocks (10–30 min for six birds, 30–120 min for one bird, 18–46 min for one bird). The total duration of the experiment, consisting of baseline, training, and probe periods, was on average 52 days. During this period, birds sang 226 (range 66–356) bouts per day during baseline days and 258 (range 171–368) bouts per day during the period of probe collection at the end of training (14% increase). The average duration of song bouts also changed little, with both the average number of target sequences per bout (8.7 during baseline, 7.7 during probes, 7% decrease) and the average number of syllables per bout (74 during baseline, 71 during probes, 2% decrease) decreasing slightly. In addition to the eight birds that completed this training paradigm, three birds were started on contextual training but never progressed to testing with probe blocks, because they did not exhibit single-context learning (n = 1); because of technical issues with consistent targeting at branch points, (n = 1); or because they lost sequence variability during initial stages of training (n = 1); these birds are excluded from the results. Of the eight birds that completed training, three birds exhibited relatively small context-dependent changes in sequencing (*Figure 1H*). We examined several variables to assess whether they could account for differences in the magnitude of learning across birds, including the bird's age, overall transition entropy of the song (*Katahira et al., 2013*), transition entropy at the targeted branch points (*Warren et al., 2012*), as well as the distance between the WN target and the closest preceding branch point in the sequence. None of these variables were significantly correlated with the degree of contextual learning that birds expressed (*Figure 4—figure supplement 1*), and consequently, all birds were treated as a single group in analysis and reporting of results. In a subset of experiments (n = 3), after completing measurements with probe blocks, we added a third, neutral context (*Figure 5*), signaled by white light, in which there was no WN reinforcement.

## Syllable sequence annotation

Syllable annotation for data analysis was performed offline. Each continuous period of singing that was separated from others by at least 2 s of silence was treated as an individual 'song' or 'song bout'. Song was bandpass filtered between 500 Hz and 10,000 Hz and segmented into syllables and

gaps based on amplitude threshold and timing parameters determined manually for each bird. A small sample of songs (approximately 20 song bouts) was then annotated manually based on visual inspection of spectrograms. These data were used to train an offline autolabeler ('hybrid-vocal-classifier', *Nicholson, 2021*), which was then used to label the remaining song bouts. Autolabeled songs were processed further in a semi-automated way depending on each bird's unique song, for example to separate or merge syllables that were not segmented correctly (detected by their duration distributions), to deal with WN covering syllables (detected by its amplitude), and to correct autolabeling errors detected based on the syllable sequence. A subset of songs was inspected manually for each bird to confirm correct labeling.

## Sequence probability analyses

Sequence probability was first calculated within each song bout as the frequency of the yellow target sequence relative to the total number of yellow and green target sequences: $p = \frac{n(target\_Y)}{n(target\_Y)+n(target\_G)}$. Note that this differs from transition probabilities at branch points in song in that it ignores possible additional syllable transitions at the branch point, and does not require the targeted sequences to be directly following the same branch point. For example for the experiment in *Figure 3*, the target sequences were 'n-ab' and 'f-ab', so the syllable covered by WN ('b' in both contexts) was two to three syllables removed from the respective branch point in the syllable sequence ('n-f' vs. 'n-a' or 'f-n' vs. 'f-a'). Note also that units of sequence probability are in percent; therefore, reported changes in percentages (e.g. *Figures 1H* and *2E,F*) describe absolute changes in sequence probability, which reflect the proportion of each target sequence, not percent changes. Song bouts that did not contain either of the two target sequences were discarded. In the plots of sequence probability over several days in *Figure 1A–C*, we calculated sequence probability for all bouts on a given day (average n = 1854 renditions of both target sequences per day). We estimated 95% confidence intervals by approximation with a normal distribution as $p \pm z * \sqrt{\frac{p*(1-p)}{n}}$ with $n = n(target\_Y) + n(target\_G)$ and z = 1.96. Context switches were processed to include only switches between adjacent blocks during the same day, that is excluding overnight switches and treating blocks as separate contexts if one day started with the same color that had been the last color on the previous day. If a bird did not produce any song during one block, this block was merged with any neighboring block of the same color (e.g. green probe without songs before green WN, where the context switch would not be noticeable for the bird). If the light color switched twice (or more) without any song bouts, those context switches were discarded.

In order to reduce variability associated with changes across individual song bouts, shift magnitude was calculated as the difference between the first five song bouts in the new context and the last five song bouts in the old context. Only context switches with at least three song bouts in each adjacent block were included in analyses of shift magnitude. In plots showing songs aligned to context switches, the x-axis is limited to show only points for which at least half of the blocks contributed data (i.e. in *Figure 2D*, half of the green probe blocks contained at least six songs). All statistical tests were performed with MATLAB. We used non-parametric tests to compare changes across birds (Wilcoxon rank-sum test for unpaired data, Wilcoxon signed-rank test for paired data), because with only eight birds/data points, it is more conservative to assume that data are not Gaussian distributed.

## Analysis of acquisition

In order to investigate how context-dependent performance developed over training (*Figure 2G–L*), we quantified changes to sequence probabilities across block switches for five birds for which we had a continuous record from the onset of training. Sequence probability curves (e.g. *Figure 2H*) for yellow switches were inverted so that both yellow and green switches were plotted in the same direction, aligned by the time of context switches, and were cut off at a time point relative to context switches where fewer than five switches contributed data. We then subtracted the mean pre-switch value from each sequence probability curve. For visual display of the example bird, sequence probability curves were smoothed with a nine bout boxcar window and displayed in bins of seven context switches. To calculate the slope of slopes and slope of intercepts (*Figure 2L*), we

calculated a linear fit to the post-switch parts of the unsmoothed sequence probability curve for each individual context switch.

## Specificity to relevant branch points

To calculate the specificity of the context difference to the targeted branch points in song, we generated transition diagrams for each bird. To simplify the diagrams, introductory notes were summarized into a single introductory state. Introductory notes were defined for each bird as up to three syllables occurring at the start of song bouts before the main motif, which tended to be quieter, more variable, with high probabilities to repeat and to transition to other introductory notes. Repeat phrases were also summarized into a single state. Motifs, or chunks, in the song with fixed order of syllables were identified by the stereotyped transitions and short gap durations between syllables in the motif (*Isola et al., 2020*; *Suge and Okanoya, 2010*) and were also summarized as a single state in the diagram. Sometimes, the same syllable can be part of several fixed chunks (*Katahira et al., 2013*), in which case it may appear several times in the transition diagram. We then calculated the difference between the transition matrices for the two probe contexts at each transition that was a branch point (defined as more than 3% and less than 97% transition probability). These context differences were split into 'targeted branch points', i.e., the branch point or branch points most closely preceding the target sequences in the two contexts, and 'non-targeted branch points', i.e., all other branch points in the song. We calculated the proportion of absolute contextual difference in the transition matrix that fell to the targeted branch points, for example for the matrix in *Figure 4C* (44 + 45)/(44 + 45 + 6+6 + 1+1 + 2+2)=83.2%. Typically, birds with clear contextual differences at the target sequence also had high specificity of sequence changes to the targeted branch points.

To calculate the transition entropy of baseline song, we again summarized introductory notes into a single introductory state. In addition, the same syllables as part of multiple fixed motifs, or in multiple positions within the same fixed motif, were renamed as different syllables, so as not to count as sequence variability what was really a stereotyped sequence (i.e. a-b 50% and b-c 50% in the fixed sequence 'abbc'). Transition entropy was then calculated as in *Katahira et al., 2013*: with x denoting the preceding syllable and y denoting the current syllable, over all syllables in the song.

## Acknowledgements

We thank Alla Karpova, Jon Sakata, Dave Mets, William Mehaffey, Assaf Breska, and Guy Avraham for helpful discussions and comments on earlier versions of this manuscript. This work was supported by the Howard Hughes Medical Institute. Lena Veit was supported as a Howard Hughes Medical Institute Fellow of the Life Sciences Research Foundation and by a postdoctoral fellowship from Leopoldina German National Academy of Sciences. Christian J Monroy Hernandez was supported by an HHMI EXROP summer fellowship.

## Additional information

### Funding

| Funder | Grant reference number | Author |
|---|---|---|
| Leopoldina German National Academy of Sciences | Postdoc Fellowship | Lena Veit |
| Life Sciences Research Foundation | Howard Hughes Medical Institute Fellowship | Lena Veit |
| Howard Hughes Medical Institute | | Michael S Brainard |
| Howard Hughes Medical Institute | EXROP summer fellowship | Christian J Monroy Hernandez |

The funders had no role in study design, data collection and interpretation, or the decision to submit the work for publication.

## Author contributions
Lena Veit, Conceptualization, Data curation, Software, Formal analysis, Supervision, Funding acquisition, Investigation, Visualization, Methodology, Writing - original draft, Writing - review and editing; Lucas Y Tian, Conceptualization, Software, Supervision, Writing - review and editing; Christian J Monroy Hernandez, Investigation, Visualization; Michael S Brainard, Conceptualization, Resources, Supervision, Funding acquisition, Writing - review and editing

## Author ORCIDs
Lena Veit (iD) https://orcid.org/0000-0002-9566-5253
Lucas Y Tian (iD) http://orcid.org/0000-0002-7346-7360
Christian J Monroy Hernandez (iD) http://orcid.org/0000-0002-3796-989X
Michael S Brainard (iD) https://orcid.org/0000-0002-9425-9907

## Ethics
Animal experimentation: All procedures were performed in accordance with protocols (#AN170723-02) approved by the University of California, San Francisco Institutional Animal Care Use Committee (IACUC).

## Decision letter and Author response
Decision letter https://doi.org/10.7554/eLife.61610.sa1
Author response https://doi.org/10.7554/eLife.61610.sa2

## Additional files

### Supplementary files
- Source code 1. Matlab code to generate *Figure 1*.
- Source code 2. Matlab code to generate *Figures 2C–F* and *3E–H*.
- Source code 3. Matlab code to generate *Figure 2L*.
- Source code 4. Matlab code to generate *Figure 4G*, *Figure 4—figure supplement 1*.
- Source code 5. Matlab code to generate *Figure 5B–D*.
- Transparent reporting form

### Data availability
Raw data are included in the manuscript and supporting files. Source data have been provided for all summary analyses, along with code to reproduce the figures.

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
