## [Decision Letter]

**Acceptance summary:**

Bengalese finches sing syntactically complex songs with flexible transitions between specific syllables. Here, the authors show that birds can modify syllable transitions depending on an arbitrary light cue. This surprising result shows that learning in the 'song system' – a neural circuit in songbirds known to drive vocal output – can be controlled by yet-to-be defined visual inputs, setting the stage for new directions in the songbird field that move beyond sequence production and into a more cognitive realm.

**Decision letter after peer review:**

Thank you for submitting your article "Songbirds can learn flexible contextual control over syllable sequencing" for consideration by *eLife*. Your article has been reviewed by 3 peer reviewers, including Jesse H Goldberg as the Reviewing Editor and Reviewer #1, and the evaluation has been overseen by Barbara Shinn-Cunningham as the Senior Editor. The following individual involved in review of your submission has agreed to reveal their identity: Constance Scharff (Reviewer #2).

The reviewers have discussed the reviews with one another and the Reviewing Editor has drafted this decision to help you prepare a revised submission.

Summary:

Veit et al. test if arbitrary visual cues can influence syllable sequence 'choices' in adult Bengalese finches. Using light- and sequence- contingent distorted auditory feedback, they find that birds robustly learn to produce context dependent syllables. The learning slowly proceeds over weeks, but once trained birds can rapidly transition between two sequence probabilities. This is a really interesting finding because it shows that the HVC chains that drive syllable phonology and sequencing can 'learn' to be gated by yet-to-be defined visual inputs. The paper is only a behavioral study without neural correlates or a candidate neural architecture that could even solve the problem – but the reviewers did not see this as a major problem for the paper. By analogy, the Tumer et al., 2007 paper from the Brainard lab was also only a behavioral study yet it has launched dozens of follow up studies and a new branch of songbird neuroscience. This paper has the potential to do the same. Follow-up studies that figure out exactly how the visual system interfaces with the songs system to dictate syllable selection will be a really interesting direction for the field – moving birdsong beyond simple sequence production and into a more cognitive realm. Thus this paper is likely to be high impact, highly cited, and important for the field.

Essential revisions:

1. Please address the question, raised by Reviewer 2, about whether or not the production of the syllable initiating the target sequence was affected by the light context. i.e. if the training was abc vs abd, did the probability of producing a or ab change depending on context? Reviwere 2 wondered, if this is indeed the case, how this would affect the interpretations of the paper, especially "postulated parallels to language?"

2. Please also address the comments below, of which there are many. A point-by-point response will not be necessary, but it should be clear that all reviewers wanted some of the claims of novelty and connection to human cognition to be tempered, i.e. keep the conclusions closer to the data. This will mostly involve re-wording in the Introduction and Discussion. There are also several sources of confusion where the methods and analyses should be presented more clearly. Please see more details below.

*Reviewer 1:*

In Figure 4: n=3 birds on Figure 4 is a bit thin. New data is not absolutely necessary because the effect is robust and consistent across birds, but the authors may want to replicate this in 1-2 more birds to really nail the finding that specific light cues can drive shifts bidirectionally.

The discussion would be improved if it included examples of natural context-dependent changes in song syllable selection – for example song matching by buntings, great tits, and sparrows. In these cases, birds sequence and select syllables in a context dependent way, even depending on where they are in a territory. Thus there may be some precedent and evolutionarily context for the core finding here, i.e. that a 'place' representation can access the song system to influence syllable sequence and song selection. This consideration would fit in the "Evolution of control over vocal sequencing" section.

*Reviewer 2:*

Line 28: 'parallels aspects of human cognitive control over speech'. I find that an overstatement, unless I misunderstand the data. The authors condition birds to avoid a particular sequence by punishing ('aversively reinforcing') it with white noise and link this to a visual stimulus. How does that parallel human cognitive control over speech? Can the authors please provide more explanation?

Line 35. Please provide a reference with evidence for the part in italics (mine) in the following statement, or rephrase more evidence-based: 'This flexibility ingrained in human language stands in striking contrast to the largely innate and stereotypic vocalization patterns of most animal species, including our closest relatives, the non-human primates.' Most of the roughly 8.7 million animal species? How many have been analyzed? Of those, how many vocalization patterns are 'largely innate'? And stereotypic? At what level stereotypic?

Line 57: '…affective behavior, elicited instinctually by contact with potential mates, rivals, or performed..' What do the authors mean by 'affective' and 'instinctually'? Human speech also has affective components (prosody for instance) and we instinctually change aspects of our speech (and language) when we talk to children, partners, strangers. This is an unsophisticated dichotomy 'humans/birdsong', please consider rephrasing (rethinking).

Line 59: 'There are differences between songs produced in distinct social contexts…' Refs to this sentence should include Heinig et al. 2014 Male mate preferences in mutual mate choice: finches modulate their songs across and within male-female interactions. Anim Behav. 97:1-12.

Line 62. ‚However, these social influences likely reflect a general modulation of song structure related to the animal's affective state (Berwick, Okanoya, Beckers, and Bolhuis, 2011). What is the concrete evidence for the 'likely' in this sentence?

Line 64 'and do not reveal whether song can be modified more flexibly by different cognitive factors.' But the fact that Bengalese finches sing different song sequences to different females (Heinig et al. paper, above) raise the possibility that 'cognitive factors' could play a role, since it's all 'affective' courtship song but different depending on which female is being sung to.

Line 78 'immediately, flexibly, and adaptively adjust their sequencing of vocal elements in response to learned contextual cues, in a manner that parallels key aspects of human cognitive control over speech' Same comment as to line 28. I think the authors are not doing themselves a favor in phrasing this claim so broadly and non-specifically.

line 110: 'alternating blocks' first mentioned here. Please include a section in the methods about blocks. I found it hard to extract the information from various points in the text how long blocks were, how many blocks per day on average (from the figures it seems that the entire day was used for data acquisition?) and what 'short probe blocks' (line 227) meant in terms of timing. Also, why were there many more block switches (Figure 1F) during baseline than during training (Figure 1G)?

Line 125: Figure 1A: it would help to point out the individual 'songs' in that figure, since song is defined differently in different species. In zebra finches a song as defined by the authors would be called 'a motif': (line 69) a 'song consist of ca 5-12 acoustically distinct elements'. Where do songs start and end? How is that determined? This relates also to my question above, whether sequences are modified in probability of occurrence or songs (or song types).

Line 237: 'Figure 3A,B shows song bouts for one example bird'. Since song bouts are defined by authors as 'separated by at least 2 sec' I would like to know whether the shown spectrogram is the entire bout and the silence before and after are just not shown or whether A and B show part of a bout. If so, can you show the entire bout, including the time when the light changes?

Line 332: 'The ability to alter syllable sequencing in a flexible fashion also contrasts with prior studies that have demonstrated modulation of vocalizations in more naturalistic settings.(…). In contrast, here we show that birds can learn to locally modulate specific features of their songs (i.e. individually targeted syllable transitions) in response to arbitrarily assigned contextual stimuli that have no prior ethological relevance.' Could the authors please comment on the following conundrum: If flexible use of song sequences under natural conditions were 'hard-wired/innate/reflexive/affective' as the authors suggest, how would the ability to pair an arbitrary contextual cue with a particular song sequence have evolved in Bengalese finches? Why would neural connections exist that allow this pairing of visual input to motor output? Isn't it more parsimonious to postulate that under natural conditions, visual stimuli do lead to different vocal motor responses because in addition to the known 'affective' mediators (hormones, dopamine etc) there is some 'top down', 'cognitive' control? (Reviewer 1 agrees with this point).

Line 348: ' Evolution of control over vocal sequencing' section is in line with my above comment, e.g. suggests what some animals might use the ability to use contextual visual information for adaptive motor output but again negates that Bengalese finches actually use it in their current behavior, instead the authors call it 'latent capacity'. I do not follow their logic.

Line 367: Neural implementation. Does the two process model relate to human speech and language? Please explain.

Line 428: add 'male'. The manuscript does not mention anywhere whether males and females in Bengalese finches sing….Or add it to line 23 in the Abstract.

Line 429: 'age range 128-320 days' was there any age-related difference in learning? Looking at the figures some birds seemed to have performed quite a bit better than others. See also line 456 below.

Line 455: 'within an interval of one to several hours' please provide more information whether this was randomly chosen or based on the birds performance. If random, what was the rationale for this large difference in block duration?

Line 456: 'after several days of training (average 33)' Please also provide the range and whether shorter training was related to age of the birds.

Line 461: 'three birds.…never progressed to full probe sequence either because they did not exhibit single-context learning or because of technical issues with consistent targeting of branch points'. Did two birds not learn and one had technical issue or other way round? How common is it in WN-escape experimental set-ups that birds do not learn? And what does 'single-context learning' mean? That they did not learn to associate yellow light with one target? This would imply that context 1 was learned first and then context 2, but in line 454 it sounds like both colors were paired with their particular target after one to several hours. Please explain.

Line 466: Please specify in the methods how many days the entire experiment lasted. How variable was the song output during that time and between individual birds? Did song output decline over time? Can the authors provide an estimate how many songs or bouts on average (and range) the birds sang?

*Reviewer 3:*

Although the present study describes the syllable sequence switching abilities of Bengalese finches within the framework of an elegantly designed behavioral paradigm, the links to the potential neural mechanisms are poorly presented or even obsolete since the authors do not provide any evidence about underlying brain dynamics. I recommend to rather discuss the results in a behavioral framework unless the authors add results from neural recordings or brain manipulations.

Line 34: Citation for reordering of finite elements to achieve infinite meaning, see Hauser, Chomsky and Fitch, 2002.

Line 47 f: Lipkind et al. 2017 showed that zebra finches can learn to re-order syllables during song learning. This paper is highly relevant and should be discussed.

Line 52: Reference to Doupe and Kuhl, 1999 should be moved to line 46?

Line 88 f: The authors decide to present most data in percentages. It would be useful to provide the actual number to assess the quality of the data.

Line 89: How reliable was the software in targeting syllables?

Line 284: The authors refer to white light as a neutral state. What is the color perception for Bengalese finches? Is white perceived rather as yellow or green? A novel light condition that the birds had not been exposed before would probably be better.

Line 306 f: Light cues are not arbitrary as the birds are initially trained to connect white noise with light of a certain color.

Line 444: What was the reason to specifically choose green and yellow as colors for this experiment?

Line 445: How much does visual perception of Bengalese finches differ between 520 and 600nm?

Line 453-455: What is the maximum duration of several hours? This would also help to understand the difference in amounts of switches in Figure 1 F and G.

Line 456: How did the white noise training look? Can learning curves be added?

Figure 1 B: It would be helpful to plot both probability curves (ab-d and ab-c) and color code them accordingly as a general probability plot (y axis).

Figure 1 F: Why is the amount of color switches different between baseline and training? When within the training did baseline days occur or was this prior to training?

Figure 1 G: Why are light phases for green/yellow differently long? The error bars are misleading, as they show the SEM of individual blocks rather than the entire sample.

---

## [Author Response]

Essential revisions:1. Please address the question, raised by Reviewer 2, about whether or not the production of the syllable initiating the target sequence was affected by the light context. i.e. if the training was abc vs abd, did the probability of producing a or ab change depending on context? Reviewer 2 wondered, if this is indeed the case, how this would affect the interpretations of the paper, especially "postulated parallels to language?"

We understand this to be a question about how specific to the target sequence were the changes in the overall transition structure of the song. We have added substantial new analysis, and a new figure (new Figure 4), to address the specificity of contextual differences to the branch points preceding the target sequences. These new analyses demonstrate that for the majority of birds, 80% or more of total contextual differences were restricted to the targeted branch points. With respect to the example cited in the question above, this means that the majority of change to sequencing in an experiment targeting ‘abc’ vs. ‘abd’ occurs to transitions at the branchpoint following syllable ‘b’ and that there is little or no contextual difference in the probability of producing an ‘a’ or ‘ab’. These new analyses indicate that the learned contextual changes to syllable sequencing reflect a capacity for modulation of specific sequences within song, rather than the kind of global modulation of structure that occurs (for example) between songs produced in different social contexts. While we have reduced comparisons with speech throughout, we note that this specificity parallels a feature of contextual modulation of sequencing in speech, which similarly reflects a capacity for flexible, local and specific reordering of elements.

2. Please also address the comments below, of which there are many. A point-by-point response will not be necessary, but it should be clear that all reviewers wanted some of the claims of novelty and connection to human cognition to be tempered, i.e. keep the conclusions closer to the data. This will mostly involve re-wording in the Introduction and Discussion. There are also several sources of confusion where the methods and analyses should be presented more clearly. Please see more details below.

We have attempted to address all comments, especially the points noted immediately above.

Reviewer 1:In Figure 4: n=3 birds on Figure 4 is a bit thin. New data is not absolutely necessary because the effect is robust and consistent across birds, but the authors may want to replicate this in 1-2 more birds to really nail the finding that specific light cues can drive shifts bidirectionally.

We have added a sentence noting that the relevant conclusions would benefit from additional experiments beyond those presented in Figure 5 (previously Figure 4).

The discussion would be improved if it included examples of natural context-dependent changes in song syllable selection – for example song matching by buntings, great tits, and sparrows. In these cases, birds sequence and select syllables in a context dependent way, even depending on where they are in a territory. Thus there may be some precedent and evolutionarily context for the core finding here, i.e. that a 'place' representation can access the song system to influence syllable sequence and song selection. This consideration would fit in the "Evolution of control over vocal sequencing" section.

We have substantially expanded the discussion of natural context-dependent changes (l.57ff, l.385ff), including addition of concrete examples, and have adjusted the logic in the Introduction and Discussion (paragraphs on evolution) to explicitly note that these examples of natural context-dependent control suggest that birds might also be able to exert such control in response to more arbitrary, learned contexts:

L.416ff: “Such capacities for vocal control likely reflect evolutionary advantages of incorporating sensory and contextual information about conspecifics and the environment in generating increasingly sophisticated vocal signaling. […] Perhaps evolutionary pressures to develop nuanced social communication led to the elaboration of cortical (pallial) control over brainstem vocal circuitry (Hage and Nieder, 2016), and thereby established a conduit that facilitated the integration of progressively more abstract cues and internal states in that control.

Reviewer 2:Line 28: 'parallels aspects of human cognitive control over speech'. I find that an overstatement, unless I misunderstand the data. The authors condition birds to avoid a particular sequence by punishing ('aversively reinforcing') it with white noise and link this to a visual stimulus. How does that parallel human cognitive control over speech? Can the authors please provide more explanation?Line 78 'immediately, flexibly, and adaptively adjust their sequencing of vocal elements in response to learned contextual cues, in a manner that parallels key aspects of human cognitive control over speech' Same comment as to line 28. I think the authors are not doing themselves a favor in phrasing this claim so broadly and non-specifically.

We have tempered, specified, or removed comparisons to contextual control of human speech throughout the text, and have provided additional explanation about similarities we see to speech control in the Discussion. We particularly focus on what we see as a shared capacity for learned, context-dependent control over the sequencing of vocal elements that is immediate, flexible, and adaptive. For example, contextual shifts appear immediately after context switches (Figure 2), they are learned, in the appropriate, arbitrarily chosen, direction in response to cues, which do not elicit such changes without prior training (Figure 1, Figure 5), and they are adaptive, in that they avoid aversive WN. We do not suggest that the context-dependent learning we have demonstrated reflects a capacity for conveying the kind of rich semantic content that is central to human language. Rather, that the underlying ability manifest in speech motor control to immediately and flexibly reorganize sequences of constituent elements (phonemes/syllables/words) to achieve a communicative ‘goal’ has some formal similarities to the simpler contextual control of vocalizations demonstrated here. In particular, we construe both to include a capacity for learned, moment-by-moment, “top-down” influences on the organization and sequencing of vocal elements to achieve contextually appropriate, adaptive outcomes. For human speech, complex cognitive processes and semantic “intent” can inform those top-down influences with an adaptive goal of influencing the listener (“conveying meaning”). For our experiments, bird vocalizations are similarly deployed in a learned and contextually appropriate fashion to achieve an adaptive goal (escaping from white noise). Correspondingly, we suggest that context-dependent modulation of vocal sequencing in the Bengalese finch may provide a particularly tractable behavioral model for examining how different arbitrary learned cues can drive the kind of top-down control of vocal motor output that forms a building block of speech. However, we appreciate the reviewer’s perspective that it is a long way from the capacities demonstrated here to insights about speech motor control, and correspondingly have largely curtailed a discussion of these parallels.

Line 35. Please provide a reference with evidence for the part in italics (mine) in the following statement, or rephrase more evidence-based: 'This flexibility ingrained in human language stands in striking contrast to the largely innate and stereotypic vocalization patterns of most animal species, including our closest relatives, the non-human primates.' Most of the roughly 8.7 million animal species? How many have been analyzed? Of those, how many vocalization patterns are 'largely innate'? And stereotypic? At what level stereotypic?

We acknowledge that the previous statement was too broad, given that most of the 8.7 million species noted by the reviewer have not been characterized in depth. We have rephrased this section (l. 37f) to focus on primates, where there has been considerable prior work:

“This cognitive control over vocal production is thought to rely on the direct innervation of brainstem and midbrain vocal networks by executive control structures in the frontal cortex, which have become more elaborate over the course of primate evolution (Hage and Nieder, 2016, Simonyan and Horwitz 2011). However, because of the comparatively limited flexibility of vocal production in nonhuman primates (Nieder and Mooney, 2020), the evolutionary and neural circuit mechanisms that have enabled the development of this flexibility remain poorly understood.“

Line 57: '…affective behavior, elicited instinctually by contact with potential mates, rivals, or performed..' What do the authors mean by 'affective' and 'instinctually'? Human speech also has affective components (prosody for instance) and we instinctually change aspects of our speech (and language) when we talk to children, partners, strangers. This is an unsophisticated dichotomy 'humans/birdsong', please consider rephrasing (rethinking).

We did not mean to imply that human speech lacks affective components. Rather, we wanted to emphasize the flexible top-down control of human speech production, which is typically considered a cognitive process involving the reorganization of vocal elements to achieve some communicative intent. In contrast, contextual changes in birdsong have typically been ascribed to affective processes, such as hormonal and neuromodulatory changes related to the production of directed song, and these changes have been shown to be unaffected by learning (“instinctual”). We here test whether cognitive influences on birdsong exist beyond these possibly completely instinctual contextual changes, building, on the previously known examples of contextual differences in birdsong. We have revised the paragraph to better explain examples of naturally occurring contextual changes in birdsong (see also other Reviewer comments), and clarified the logic in Introduction and Discussion.

Line 59: 'There are differences between songs produced in distinct social contexts…' Refs to this sentence should include Heinig et al. 2014 Male mate preferences in mutual mate choice: finches modulate their songs across and within male-female interactions. Anim Behav. 97:1-12.

We have added the reference.

Line 62. ‚However, these social influences likely reflect a general modulation of song structure related to the animal's affective state (Berwick, Okanoya, Beckers, and Bolhuis, 2011). What is the concrete evidence for the 'likely' in this sentence?

Changes in directed song typically reflect a general or global modulation of song structure, such that song overall is faster, louder, higher pitched, and more stereotyped. We have rephrased and included references in the paragraph to clarify.

l. 57ff: “Contextual variation of song in natural settings, such as territorial counter-singing or female-directed courtship song, indicate that songbirds can rapidly alter aspects of their song, including syllable sequencing and selection of song types (Chen, Matheson, and Sakata, 2016; Heinig et al., 2014; King and McGregor, 2016; Sakata, Hampton, and Brainard, 2008; Searcy and Beecher, 2009; Trillo and Vehrencamp, 2005). […] For example, the presence of potential mates or rivals elicits a global and unlearned modulation of song intensity (James, Dai, and Sakata, 2018a) related to the singer’s level of arousal or aggression (Alcami, Ma, and Gahr, 2021; Heinig et al., 2014; Jaffe and Brainard, 2020).”

Line 64 'and do not reveal whether song can be modified more flexibly by different cognitive factors.' But the fact that Bengalese finches sing different song sequences to different females (Heinig et al. paper, above) raise the possibility that 'cognitive factors' could play a role, since it's all 'affective' courtship song but different depending on which female is being sung to.

To our understanding, the Heinig et al. paper shows that birds sing different *intensity* of directed song to different females, which is compatible with an interpretation that they have different levels of general motivation to sing directed song to different females. This and other papers about directed song, referenced in response to the previous comment, suggest that directed song can elicit global changes (increased speed, amplitude, stereotypy, including sequence stereotypy) that are not learned. In contrast, the changes we show here are learned in response to arbitrary contextual cues, and are specific to the targeted position in the song bout. We have added new analysis (Figure 4) that demonstrates this specificity of the contextual changes in the current experiment, and have also clarified logic in Introduction and Discussion, to note that song changes elicited in natural contexts – including song type selection in other species – raise the possibility that learned, cognitive factors could play a role in modulating vocal output, an idea that we attempt to specifically test in our study.

line 110: 'alternating blocks' first mentioned here. Please include a section in the methods about blocks. I found it hard to extract the information from various points in the text how long blocks were, how many blocks per day on average (from the figures it seems that the entire day was used for data acquisition?) and what 'short probe blocks' (line 227) meant in terms of timing. Also, why were there many more block switches (Figure 1F) during baseline than during training (Figure 1G)?Line 455: 'within an interval of one to several hours' please provide more information whether this was randomly chosen or based on the birds performance. If random, what was the rationale for this large difference in block duration?

These details have been added to the “*Training procedure and blocks”* section of the Methods.

Line 125: Figure 1A: it would help to point out the individual 'songs' in that figure, since song is defined differently in different species. In zebra finches a song as defined by the authors would be called 'a motif': (line 69) a 'song consist of ca 5-12 acoustically distinct elements'. Where do songs start and end? How is that determined? This relates also to my question above, whether sequences are modified in probability of occurrence or songs (or song types).

We believe these questions are mainly based on a misunderstanding of the referenced sentence in l.72. For clarification, we have rephrased as follows: “Each Bengalese finch song repertoire included ~5-12 acoustically distinct elements (‘syllables’) that are strung together into long sequences in variable but non-random order”. A song, or song bout, (which we colloquially use interchangeably but have now tried to exclusively use song bout in the manuscript) is defined as is typical in the field, for zebra finch and Bengalese finch, as a period of continuous vocalizations separated by 2s of silence (now defined in the Methods). There are no different song types in Bengalese finches, and each song bout typically contains several renditions of both target sequences.

Line 237: 'Figure 3A,B shows song bouts for one example bird'. Since song bouts are defined by authors as 'separated by at least 2 sec' I would like to know whether the shown spectrogram is the entire bout and the silence before and after are just not shown or whether A and B show part of a bout. If so, can you show the entire bout, including the time when the light changes?

The spectrograms do not show the entire bout, but show an exemplary section of the bout, to make it easier to recognize target sequences. We have added a supplementary to Figure 3 to show the entire bout. We cannot show the time of the light change, as the recording program was set up to never change lights in the middle of song, i.e. the light changed as soon as the recording for the first bout ended, and before the recording for any following bout started.

Line 332: 'The ability to alter syllable sequencing in a flexible fashion also contrasts with prior studies that have demonstrated modulation of vocalizations in more naturalistic settings.(…). In contrast, here we show that birds can learn to locally modulate specific features of their songs (i.e. individually targeted syllable transitions) in response to arbitrarily assigned contextual stimuli that have no prior ethological relevance.' Could the authors please comment on the following conundrum: If flexible use of song sequences under natural conditions were 'hard-wired/innate/reflexive/affective' as the authors suggest, how would the ability to pair an arbitrary contextual cue with a particular song sequence have evolved in Bengalese finches? Why would neural connections exist that allow this pairing of visual input to motor output? Isn't it more parsimonious to postulate that under natural conditions, visual stimuli do lead to different vocal motor responses because in addition to the known 'affective' mediators (hormones, dopamine etc) there is some 'top down', 'cognitive' control? (Reviewer 1 agrees with this point).Line 348: ' Evolution of control over vocal sequencing' section is in line with my above comment, e.g. suggests what some animals might use the ability to use contextual visual information for adaptive motor output but again negates that Bengalese finches actually use it in their current behavior, instead the authors call it 'latent capacity'. I do not follow their logic.

We largely agree with this interpretation and have now added discussion to clarify this logic in the “Evolution” paragraph, and explicitly state that some examples of natural contextual variation likely also involve more cognitive processing.

l. 420f: “and suggest that some of the contextual control observed in natural settings may likewise rely on learned associations and other cognitive factors.”

Line 367: Neural implementation. Does the two process model relate to human speech and language? Please explain.

We have added references to speech motor adaptation and language selection studies, and clarified that the rest of this paragraph concerns models related to more general motor control processes.

Line 428: add 'male'. The manuscript does not mention anywhere whether males and females in Bengalese finches sing….Or add it to line 23 in the Abstract.

Done.

Line 429: 'age range 128-320 days' was there any age-related difference in learning? Looking at the figures some birds seemed to have performed quite a bit better than others. See also line 456 below.Line 456: 'after several days of training (average 33)' Please also provide the range and whether shorter training was related to age of the birds.

We have added analyses in Sup. Figure 4 to examine whether the bird’s performance depended on age and other possible explanatory variables. We did not find a significant correlation with any tested variables, although that may be expected given the small sample size and idiosyncratic features of each song and choice of branch point. Follow-up studies would need to be performed which systematically test learning ability at different branch points of the same bird.

We now also note that training duration (range 12-79 days) was not varied across birds in a fashion that was explicitly related to magnitude of sequence changes, and indeed was not a tightly controlled variable in these experiments.

Line 461: 'three birds.…never progressed to full probe sequence either because they did not exhibit single-context learning or because of technical issues with consistent targeting of branch points'. Did two birds not learn and one had technical issue or other way round? How common is it in WN-escape experimental set-ups that birds do not learn? And what does 'single-context learning' mean? That they did not learn to associate yellow light with one target? This would imply that context 1 was learned first and then context 2, but in line 454 it sounds like both colors were paired with their particular target after one to several hours. Please explain.

We have clarified that one bird did not learn in single context training, one bird was abandoned due to technical difficulties, and one bird exhibited a loss of sequence variability during initial training that prevented further differential training.

We first tested each bird to ensure that it was capable of “single context learning” before initiating context dependent training. Single context learning as now defined in results means learning in one direction with constant light color, as in Figure 1C,D, and we construed this as a likely pre-requisite for context-dependent learning. It happens sometimes that birds do not learn in WN-escape experiments, typically because of some higher-order structure in the song (such as history dependence as described by Warren et al. 2012). However, for one pilot bird that did not exhibit single context learning in these initial experiments (similar to 1C,D) we nonetheless initiated dual context-dependent training to see if it might develop learning over time. We never saw evidence of learning in that bird, and training was abandoned.

A second bird was excluded because of technical issues with maintaining accurate targeting of syllables through template matching (see Methods) to deliver WN.

The third excluded bird learned well in single context training, but lost sequence variability over the course of initial training for reasons that are unclear and in a manner that was not observed in other birds. This resulted in the elimination of one of the target sequences, precluding differential training, and the bird was abandoned. These further details are now noted in Methods.

Line 466: Please specify in the methods how many days the entire experiment lasted. How variable was the song output during that time and between individual birds? Did song output decline over time? Can the authors provide an estimate how many songs or bouts on average (and range) the birds sang?

Consistent with prior observations (Yamahachi et al., 2020 Plos One), we found that some birds increased and others decreased the average number of song bouts per day. On average, birds sang 226 (range 66-356) bouts during baseline days and 258 (range 171-368) bouts per day during the period of probe collection at the end of training (14% increase). The average duration of song bouts also changed little, with both the average number of target sequences per bout (8.7 during baseline, 7.7 during probes, 7% decrease) and the average number of syllables per bout (74 during baseline, 71 during probes, 2% decrease) decreasing slightly. These numbers are now included in Methods.

**Author response image 2. respfig2:** 

Reviewer 3:Although the present study describes the syllable sequence switching abilities of Bengalese finches within the framework of an elegantly designed behavioral paradigm, the links to the potential neural mechanisms are poorly presented or even obsolete since the authors do not provide any evidence about underlying brain dynamics. I recommend to rather discuss the results in a behavioral framework unless the authors add results from neural recordings or brain manipulations.

We have expanded Introduction and Discussion on behavioral studies in birds and humans, and have reserved any speculation about neural mechanisms for the discussion. We retained some discussion of this point, as songbirds are an extensively studied model for neural mechanisms of vocal motor control; this large prior body of work on neural mechanisms enables some informed speculation about how the ability to rapidly adjust song in response to learned, visual cues could be accomplished by the song system, and we felt this would be of potential interest to more mechanistically inclined readers.

Line 34: Citation for reordering of finite elements to achieve infinite meaning, see Hauser, Chomsky and Fitch, 2002Line 47 f: Lipkind et al. 2017 showed that zebra finches can learn to re-order syllables during song learning. This paper is highly relevant and should be discussed.Line 52: Reference to Doupe and Kuhl, 1999 should be moved to line 46?

We have added these references.

Line 88 f: The authors decide to present most data in percentages. It would be useful to provide the actual number to assess the quality of the data.

We have clarified that the measure used throughout this study, sequence probability, is in units of percent. The changes that we describe in percentages are absolute changes in sequence probability, which reflect the proportion of each target sequence, not percent changes. For example, the plots in Figure 2 A-D (and similar ones throughout the manuscript) show raw, absolute values of sequence probability. To provide a measure of the quality of the data, we provide error bars, reflecting s.e.m. across song bouts.

We previously had not provided error bars on Figure 1 B-D, as these data points are based on all target sequences sung on a given day (i.e. there is only one data point per day). We have now added confidence intervals to Figure 1 B-D, estimated from normal approximation of the binomial probability of the proportion of ‘abd’ and ‘abc’ target sequences. The actual number of either target sequence are 1854 per day during baseline, 808 per day during ab-d targeting (fewer, because day 1 and day 4 are not full days of singing, but belong partly to baseline or ab-c targeting), 1888 per day during ab-c targeting.

Line 89: How reliable was the software in targeting syllables?

The reliability of the templates was checked continuously throughout the experiment, but we did not keep careful notes on this for each bird. We have therefore retroactively assessed the specificity and sensitivity of the template by hand-checking 20 randomly selected song bouts from a single day of training for each bird, and added this information to Methods:

“We determined sensitivity and specificity of the template to the target sequence on a randomly selected set of 20 song bouts on which labels and delivery of WN was hand-checked. […] The average template specificity was 96.7% (range 90.6-100% ).”

Line 284: The authors refer to white light as a neutral state. What is the color perception for Bengalese finches? Is white perceived rather as yellow or green? A novel light condition that the birds had not been exposed before would probably be better.Line 306 f: Light cues are not arbitrary as the birds are initially trained to connect white noise with light of a certain color.Line 444: What was the reason to specifically choose green and yellow as colors for this experiment?Line 445: How much does visual perception of Bengalese finches differ between 520 and 600nm?

We mean by ‘arbitrary’ that these colors have no prior ethological meaning for the behavior of the bird. Hence, the color perception should not matter, as long as the birds are able to discriminate the colors. We set out to demonstrate that the birds would be able to associate song changes with any arbitrary visual stimulus. There was no reason to choose these specific colors. The white light is “neutral” only insofar as the birds had learned that aversive WN would never occur in the white context, not because it is spectrally in the middle of green and yellow. We did use a white LED light which was different from the home light in the cage, which was also white and the birds might have had experience with prior to any context training.

Line 453-455: What is the maximum duration of several hours? This would also help to understand the difference in amounts of switches in Figure 1 F and G.Figure 1 F: Why is the amount of color switches different between baseline and training? When within the training did baseline days occur or was this prior to training?Figure 1 G: Why are light phases for green/yellow differently long? The error bars are misleading, as they show the SEM of individual blocks rather than the entire sample.

We have added further explanation of block durations in the Methods (see also Reviewer2). Probes were collected before WN training. The training schedule was changed between 1F and 1G. The individual light phases are drawn from random intervals, therefore, it might randomly happen that on one day the yellow contexts appear longer than the green context (or vice versa), but the two colors are drawn from the same intervals, so this should even out over time. We think that the SEM per block should be informative about effect reliability, and have expanded the legend for Figure 1G so as to clarify this measure.

Line 456: How did the white noise training look? Can learning curves be added?

Learning data are shown in Figure 1 C and D for single context training, and Figure 2 G,H for the contextual training protocol.

Figure 1 B: It would be helpful to plot both probability curves (ab-d and ab-c) and color code them accordingly as a general probability plot (y axis).

We now have added the probability for the other target sequence; thank you for the suggestion.